# MRI-based deep learning can discriminate between temporal lobe epilepsy, Alzheimer's disease, and healthy controls

Allen J. Chang [1], Rebecca Roth[2], Eleni Bougioukli[1], Theodor Ruber [3], Simon S. Keller[4], Daniel L. Drane[2], Robert E. Gross[5], James Welsh[1], Anees Abrol [6], Vince Calhoun [7], Ioannis Karakis[2], Erik Kaestner[8], Bernd Weber[9], Carrie McDonald[8], Ezequiel Gleichgerrcht[1,10], Leonardo Bonilha[2,10 ✉] & Alzheimer's Disease Neuroimaging Initiative*

## Abstract

**Background** Radiological identification of temporal lobe epilepsy (TLE) is crucial for diagnosis and treatment planning. TLE neuroimaging abnormalities are pervasive at the group level, but they can be subtle and difficult to identify by visual inspection of individual scans, prompting applications of artificial intelligence (AI) assisted technologies.

**Method** We assessed the ability of a convolutional neural network (CNN) algorithm to classify TLE vs. patients with AD vs. healthy controls using T1-weighted magnetic resonance imaging (MRI) scans. We used feature visualization techniques to identify regions the CNN employed to differentiate disease types.

**Results** We show the following classification results: healthy control accuracy = 81.54% (SD = 1.77%), precision = 0.81 (SD = 0.02), recall = 0.85 (SD = 0.03), and F1-score = 0.83 (SD = 0.02); TLE accuracy = 90.45% (SD = 1.59%), precision = 0.86 (SD = 0.03), recall = 0.86 (SD = 0.04), and F1-score = 0.85 (SD = 0.04); and AD accuracy = 88.52% (SD = 1.27%), precision = 0.64 (SD = 0.05), recall = 0.53 (SD = 0.07), and F1 score = 0.58 (0.05). The high accuracy in identification of TLE was remarkable, considering that only 47% of the cohort had deemed to be lesional based on MRI alone. Model predictions were also considerably better than random permutation classifications ($p < 0.01$) and were independent of age effects.

**Conclusions** AI (CNN deep learning) can classify and distinguish TLE, underscoring its potential utility for future computer-aided radiological assessments of epilepsy, especially for patients who do not exhibit easily identifiable TLE associated MRI features (e.g., hippocampal sclerosis).

## Plain language summary

In people with temporal lobe epilepsy, seizures start in a particular part of the brain positioned behind the ears called the temporal lobe. It is difficult for a doctor to detect that a person has temporal lobe epilepsy using brain scans. In this study, we developed a computer model that was able to identify people with temporal lobe epilepsy from scans of their brain. This computer model could be used to help doctors identify temporal lobe epilepsy from brain scans in the future.

[1] Department of Neurology, Medical University of South Carolina, Charleston, SC, USA. [2] Department of Neurology, Emory University School of Medicine, Atlanta, GA, USA. [3] Department of Epileptology, University Hospital Bonn, Venusberg-Campus 1, 53127 Bonn, Germany. [4] Department of Pharmacology and Therapeutics, Institute of Systems, Molecular and Integrative Biology, University of Liverpool, Liverpool, UK. [5] Department of Neurosurgery, Emory University Hospital, Atlanta, GA, USA. [6] Tri-institutional Center for Translational Research in Neuroimaging and Data Science (TReNDS), Georgia State University, Georgia Institute of Technology, Emory University, Atlanta, GA, USA. [7] School of Electrical & Computer Engineering, Georgia Institute of Technology, Atlanta, GA, USA. [8] Department of Psychology, University of California, San Diego, CA, USA. [9] Institute of Experimental Epileptology and Cognition Research, University Hospital Bonn, Venusberg-Campus 1, 53127 Bonn, Germany. [10]These authors contributed equally: Ezequiel Gleichgerrcht, Leonardo Bonilha. *A list of authors and their affiliations appears at the end of the paper. ✉email: leonardo.bonilha@emory.edu

Temporal lobe epilepsy (TLE) is defined by seizures originating in the temporal lobe[1] and remains the most common form of medication-refractory epilepsy in adults[2]. The mainstay of TLE diagnosis is the identification of ictal EEG onset patterns localized to the temporal lobe during seizure-monitoring[3]. Neuroradiological identification of a temporal lobe lesion is not required for diagnosis per se but constitutes powerful confirmatory evidence of TLE. The majority of TLE cases are associated with medial temporal sclerosis (MTS), which is a histological abnormality defined by cell loss and gliosis in the hippocampus and surrounding medial temporal structures. Signs associated with MTS can be detected during visual inspection of pre-surgical MRI and often present as hippocampal atrophy on T1-weighted images[4], which is linearly associated with hippocampal cell loss[5], and increased signal on T2-weighted images[6–8], reflecting underlying gliosis. However, in many cases, MTS is not visually apparent on MRI and is only confirmed by histopathological analysis of surgical specimens[9].

The identification of neuroimaging signs indicative of TLE is crucial during diagnostic evaluation since the presence of MRI abnormalities is commonly associated with a greater likelihood of surgical success[10–22]. Therefore, strategies to improve neuroimaging diagnostic accuracy are of great importance and can have a direct impact on the investigation and treatment of epilepsies.

An interesting aspect of brain atrophy in TLE is that it follows a consistent pattern within specific areas of gray matter tissue but can vary across individuals (Fig. 1). Conceptually, as far as computer-vision is concerned, this is comparable to a handwritten digit, which may vary depending on the writer, but also follows a consistent pattern for each digit. As such, modern techniques of computer vision, more specifically, convolutional neural networks (CNN) used for image classification, could be directly employed to address this problem.

In this study, we hypothesized that 2D CNN optimized for TLE classification could leverage TLE whole brain atrophy patterns for disease classification. Thus, we applied CNN to modern approaches for T1-weighted voxel-based quantification of tissue integrity and tested the classification accuracy of CNN to detect and discriminate TLE from healthy controls. Importantly, other neurological diseases, notably Alzheimer's disease (AD), can also be associated with gray matter atrophy involving the medial temporal lobe and the limbic system[23]. It is therefore paramount to test whether deep learning can not only detect the presence of abnormalities, but discriminate TLE-specific patterns from other forms of non-specific brain injury or atrophy in similar brain regions.

Finally, we investigated the regional anatomical feature importance in classification. We hypothesized that CNN would classify TLE with accuracy higher than chance and that medial temporal and limbic structures would be of high feature importance in classification. Our CNN models had a mean accuracy of 86.84% (SD = 1.33%), mean precision of 0.77 (SD = 0.03), a mean recall of 0.74 (SD = 0.03), and an F1-score of 0.75 (SD = 0.025) for disease prediction. Feature analysis of these models revealed temporal as well as extratemporal regions, such as those found in the frontal (e.g., orbital, and olfactory) and occipital cortices (e.g., precuneus), to be of high importance.

## Methods

**Data sources**. Participants with TLE and their matched healthy controls were derived from three different sites: The Medical University of South Carolina (Charleston, SC, USA), Emory University (Atlanta, GA, USA), and The University of Bonn (Bonn, Germany). Patients were recruited sequentially between March 2017 and December 2020 if they met the following inclusion criteria: (1) a diagnosis of drug-resistant unilateral temporal lobe epilepsy was achieved by the treating clinical team based on a combination of clinical, neurophysiological, radiographic, and neuropsychological findings in accordance with criteria set for by the International League Against Epilepsy (ILAE)[1]; (2) treatment with either resection or laser interstitial thermoablation as decided by the patient and their clinical team; (3) age > 18 years old. Patients were excluded if they had mass occupying lesions (e.g., tumors, vascular malformations), as these tend to distort the anatomy, if they did not undergo resective/ablative surgery, or if they were found to have bilateral temporal lobe epilepsy or an additional extratemporal focus. For patients with radiographic findings of underlying hippocampal sclerosis (47% of the sample), a diagnosis was further confirmed if seizure semiology and scalp EEG patterns matched those expected on the side of radiographic changes. Often, patients required invasive EEG monitoring to confirm the unilateral medial temporal onset of seizures, particularly if (a) they were deemed non-lesional on MRI, (b) had poorly localized or lateralized seizures on scalp EEG, and/or (c) had discordant findings. Frequently, additional neuroimaging supplemented the evidence favoring medial temporal ictal onset, particularly PET and SISCOM SPECT analysis. The final diagnosis was achieved by consensus of surgical conferences at each center. We did not include other causes of lesional temporal epilepsy such as tumors (e.g., DNET) or vascular malformations (e.g., cavernoma) to first, avoid anatomical biases in the sample, and second, such lesions tend be to be readily detected by human visual inspection. All relevant ethical regulations were followed, and informed consent was obtained at the respective facilities. IRB approval was obtained through the Institutional Review Board where participants were enrolled, these were the Medical University of South Carolina, Emory University, and University of Bonn.

Additional data used in the preparation of this article were obtained and approved from the Alzheimer's Disease Neuroimaging Initiative (ADNI) database (adni.loni.usc.edu). The ADNI was launched in 2003 as a public-private partnership, led by Principal Investigator Michael W. Weiner, MD. The primary goal of ADNI has been to test whether serial magnetic resonance imaging (MRI), positron emission tomography (PET), other biological markers, and clinical and neuropsychological assessment can be combined to measure the progression of mild cognitive impairment (MCI) and early Alzheimer's disease (AD).

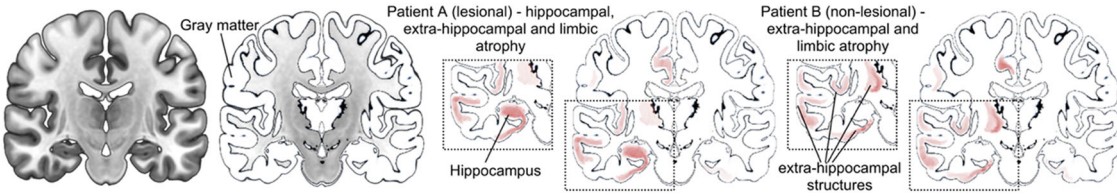

**Fig. 1 Conceptual representation of brain atrophy in two hypothetical patients with TLE.** Patient A is an example of a lesional case and Patient B is an example of a non-lesional case. Note that atrophy (red regions) may occur in different regions and with subtle differences in intensity, but the pattern of atrophy follows a similar extra-hippocampal and limbic anatomical distribution.

From this database, we collected T1-weighted images and demographic information of participants with Alzheimer disease (AD) as well as matched healthy controls. AD patients were diagnosed based on ADNI inclusion criteria, which include a probable AD diagnosis based on the National Institute on Aging-Alzheimer's Association guidelines for the neuropathological assessment of AD. ADNI excludes any patients with AD that may also have comorbid epilepsy, among other neurologic disorders. Similarly, there were no cases in our sample of patients with a diagnosis of TLE who also met criteria for AD or other neurodegenerative disorders. ADNI participants gave informed consent and IRB approval was obtained at each participating location.

**Participants**. The study consisted of a total of 157 participants diagnosed with medial TLE, 73 participants diagnosed with early AD, and 251 healthy controls (150 derived from the epilepsy age-matched controls and 101 derived from the ADNI database). Ninety-two of participants had an ictal onset zone in the left temporal lobe, 65 had an ictal onset zone in the right temporal lobe, and 1 had bilateral onset. TLE participants were on average 38.68 years old (SD = 12.45) and 60% female (40% male). AD participants were on average 75.71 years old (SD = 8.10) and 60.3% male (39.7% female). Healthy participants were on average 51.59 years old (SD = 20.97) and 51% male (49% female). Demographic summary is displayed in Table 1. Epilepsy participants had an average age of onset of 17.05 years (SD = 12.39) and duration of 21.62 years (SD = 14.84).

**MRI acquisition**. The images for TLE patients were obtained preoperatively on a 3T MRI scanner at each site. Scanner type and acquisition parameters varied across institutions. MUSC: Siemens Skyra 3T scanner, isotropic voxel size 1 mm, 12-channel head coil, TR = 2050–2250 ms, TE = 2.5–18 ms, FOV = 256–320 mm, flip angle 10°. Emory: Siemens Prisma 3T scanner, isotropic voxel size 0.8 mm, 12-channel head coil, TR = 2300 ms, TE = 2.75 ms, TI = 1100 ms, flip angle 8. Bonn: Siemens Magnetom Trio 3T scanner, 8-channel head coil, isotropic voxel size of 1 mm, TR = 650 ms, TE = 3.97 ms, TI = 650 ms, flip angle 10°.

All T1-weighted images from AD participants and their HC were downloaded from the ADNI database. ADNI data were acquired across a variety of scanners with protocols individualized for each scanner. An example protocol for a MRI system (Magnetom Sonata Syngo; Siemens Medical Solutions, Malvern, PA), running version MR 2004A software, is the sagittal inversion-prepared three-dimensional T1-weighted gradient-echo sequence (magnetization-prepared rigid acquisition gradient echo or equivalent), with the following parameters: repetition time (TR) 2400 ms; echo time (TE) 3.5 ms; inversion time (TI) 1000 ms; flip angle 8 degrees; bandwidth 180 Hz/pixel; field of view (FOV) 240 mm; matrix 192 192; number of slices 60; slice thickness 1.2 mm.

**Table 1 Demographic summary.**

| Disease | N sample | Age (SD[a]) | Sex (F/M[b]) |
|---|---|---|---|
| Temporal lobe epilepsy | 157 | 38.77 (12.44) | 97/62 |
| Alzheimer's disease | 73 | 75.71 (8.10) | 29/44 |
| Healthy | | | |
| ADNI[3] | 101 | 73.48 (6.55) | 50/51 |
| Epilepsy-age matched | 150 | 36.75 (12.74) | 73/75 |
| Total | 481 | 51.01 (20.74) | 250/232 |

[a]Standard deviation.
[b]Female/male.
[c]Alzheimers Disease Neuroimaging Initiative.

**Preprocessing**. Once T1-weighted images were collected and allocated to either training, validation, or testing sets, we preprocessed images for input into the CNN (see section on data allocation below). This preprocessing pipeline included: normalization, tissue segmentation, smoothing/thresholding, slice extraction, and labeling. The process of normalization, tissue segmentation, and smoothing/thresholding is displayed in Fig. 2.

**MRI normalization**. We normalized all T1-weighed images into standard stereotaxic MNI152 space (113 × 137 × 113) using the normalize function from the software package Statistical Parametric Mapping (SPM) with the following parameters: bias regularization = 0.0001, bias FWHM = 60, tissue probability map = TPM.nii, voxel size = $1 \times 1 \times 1$ mm$^3$, and 4th degree b-spline interpolation.

**Tissue segmentation, smoothing, and thresholding**. We used SPM with the CAT12 extension toolbox to segment brain tissues with the default parameters. Following tissue segmentation, we smoothed gray and white segmented images using SPM's smooth function (a three-dimensional FWHM, 10 mm). The smoothing was performed to minimize individual variability in sulci and gyri positioning. Only voxels with more than 20% of probability of being gray matter were included in the analyses.

**Slice extraction and labeling**. We extracted the 58 middle axial slices (−29 to +28 mm) from each participant's smoothed and normalized gray matter images. We limited axial slice extraction to those between −29 and +28 mm to avoid overfitting by eliminating slices that would have low classification relevance in the model, i.e., inputting axial slices at the superior or inferior extremes that would contain fewer voxels with classification weight. We then labeled each image according to the corresponding diagnosis. Thus, we produced 58 labeled images per participant.

**Voxel-wise linear age regression**. Since our dataset is not age balanced, we removed potential age-related effects on gray matter by performing a voxel-wise linear regression transformation. We fitted a linear relationship between age and voxel intensity for each voxel using data from all participants. We then replaced the value in each voxel for each participant with the residual value from the linear relationship between age and gray matter value. An illustrative example of this transformation is shown in Fig. 3.

**Training/validation/testing set allocation**. We randomly divided participants from each group (i.e., Alzheimer's, TLE, and healthy controls) into three cohorts: (1) Training, (2), Testing, and (3) Validation sets. The training group comprised 60% of the participants and was used to train the model. The testing set comprised 25% of the participants and was used to test the model. The validation set was composed by the remaining 15% of the participants and was used to optimize training of the model. For example, the TLE dataset consists of 157 participants, who were randomly assigned into three groups with 94 TLE participants allocated to the training group (~60% of 157), 39 TLE participants assigned to the testing group (~25% of 157), and the remaining 25 TLE participants allocated to the validation group (~15% of 157).A Randomized group allocation was performed before each model construction and evaluation using MATLAB's "dividerand" function (https://www.mathworks.com/help/deeplearning/ref/dividerand.html).

**Artificial neural network architecture**. We used a two-dimensional convolutional neural network (CNN) architecture

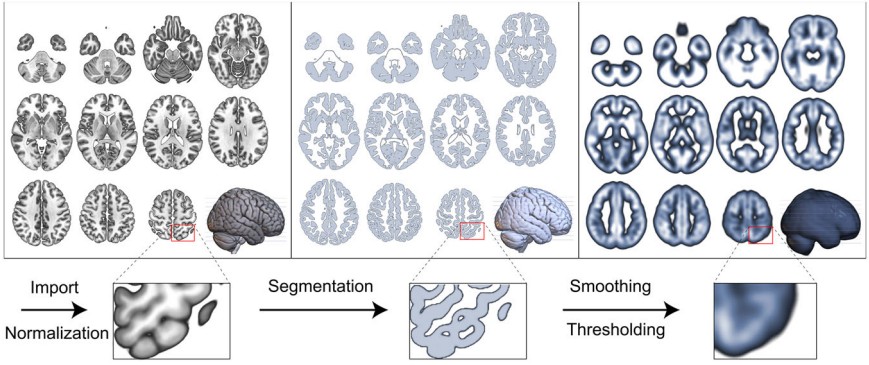

**Fig. 2 Image preprocessing pipeline.** Image preprocessing pipeline was used to normalize the images to the MNI152 standard space, segment the brain into probabilistic maps of gray matter, which were subsequently smoothed to reduce interindividual sulci and gyri positioning.

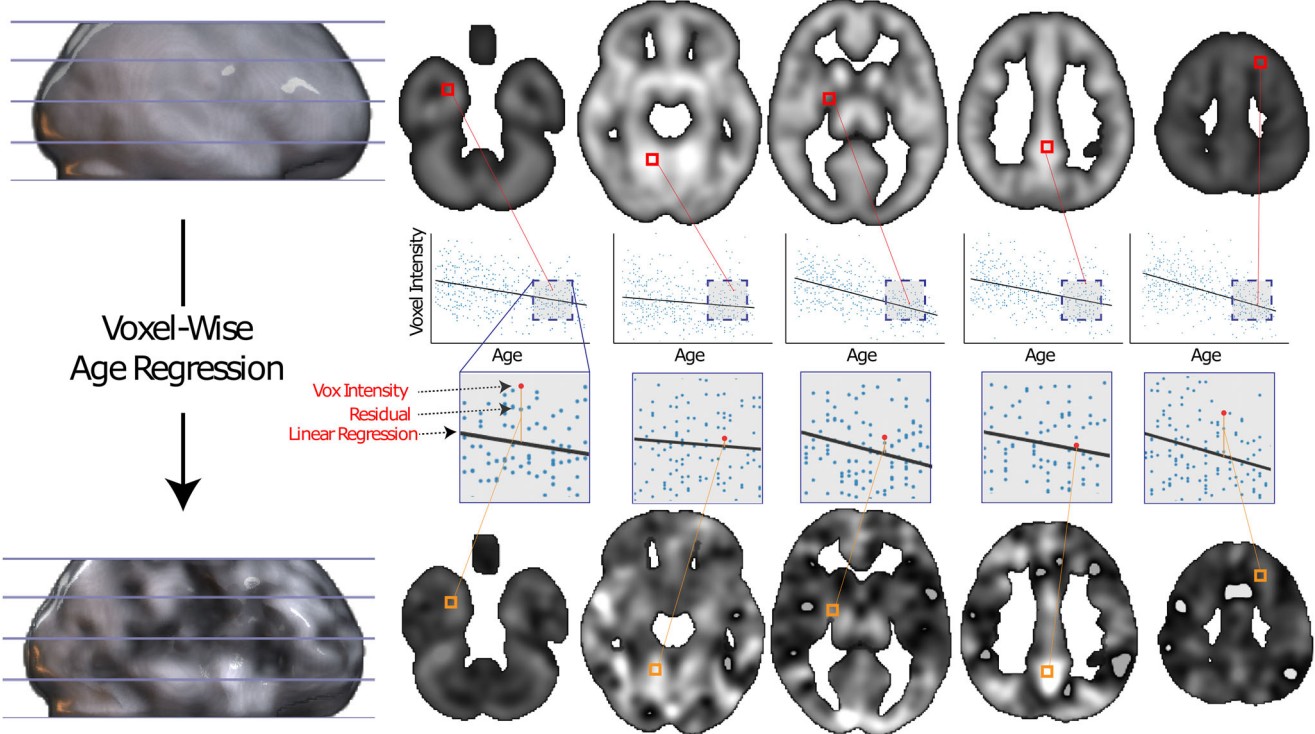

**Fig. 3 An example of our voxel-wise age regression method.** Red inset indicates the original voxel characteristics including intensity and location. Linear regression lines represent the linear fit between age and voxel gray matter from participants. Orange insets indicate the regression residual, which is used as the new transformed age regressed voxel intensity.

for our artificial neural network design. Our CNN contained three main layers: (1) an input layer, (2) hidden layer(s), and (3) an output layer. Our input layer consisted of a $113 \times 137 \times 1$ activations (i.e., the first two dimensions of an axial slice in the MNI space plus the third dimension as the grayscale channel). Our hidden layers contained three convolutional modules, each consisting of four components: (a) a $3 \times 3$ convolutional component with a [1 1] stride, (b) a batch normalization component, (c) a ReLU component, and d) a max pooling component. These three modules were followed by a classification layer (i.e., two fully connected layers followed by a softmax). The output layer contained the three disease classes (i.e., TLE, AD, and Healthy controls). Figure 4 depicts the network's architecture in more detail.

**Model execution.** To determine the stability of the algorithm, we created one hundred models with different permutations of the training, testing, and validation groups derived from the same datasets at the same ratios to one another (i.e., 60% training, 15% validation, and 25% testing). This produced a distribution of algorithm performances, allowing us to examine the overall stability of the algorithm.

We used MATLAB's Deep Learning Toolbox to construct and execute the CNN. To optimize the objective function, we used a stochastic gradient descent with momentum (SGDM) optimizer with an initial learning rate of 0.01 (default value for SGDM in MATLAB). Max number of epochs for training was set to thirty with the training and validation set being shuffled before each epoch. The remaining settings were set to default and included the following: MiniBatchSize = 128, ValidationFrequency = 50, ValidationPatience = Inf, LearnRateSchedule = 'none', L2Regularization = 0.0001, GradientDecayFactor = 0.9, SquaredDecayFactor = 0.9, Epsilon = $10^{-8}$, ResetInputNormalization = true, BatchNormalizationStatistics = 'population'.

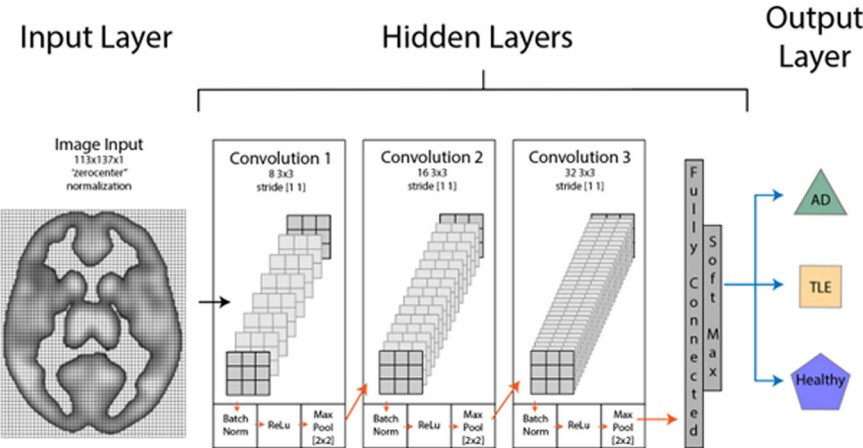

**Fig. 4 Convolutional network architecture.** Layers are linked in-series with the input layer connected to the hidden layers which is connected to the output layer. Hidden layers are comprised of three convolutional units (convolution, batch norm, ReLu, and max pool) and a classification layer (fully connected, and soft max).

**Model evaluation**. To evaluate the performance of the trained models, we compared the trained model's accuracy distribution to that of a model trained on improperly labeled data (i.e., we shuffled the labels for the training dataset while testing and validation labels remained accurate). From our models trained on the improperly labeled training dataset (hereby known as "shuffled" models), we were able to evaluate our properly trained model's performance by obtaining a frequency distribution comparison index (FDCI) value, calculated with the following formula:

$$\frac{\# \text{ of times model } A's \text{ accuracy is higher than model } B's \text{ accuracy}}{\# \text{ of comparisons}}. \quad (1)$$

where model A was our properly trained model, and model B is our improperly trained shuffled model. From this formula, we obtained a number from 0 to 1 where 0 indicates that model A never outperformed model B, and 1 indicates that model A always outperformed model B. This technique has been applied similarly to a recently published paper that used deep learning in participants with epilepsy[24]. To evaluate the stability of the algorithm, we created 200 models (100 for the properly trained, 100 for shuffled) to obtain a distribution of model performances.

The model was tasked to predict slice level disease diagnosis, the following metrics were reported at the slice level:

Model accuracy was calculated using the following formula:

$$\frac{True\ Postive + True\ Negative}{Total\ Comparisons}. \quad (2)$$

Model precision was calculated using the following formula:

$$\frac{True\ Positive}{True\ Positive + False\ Positive}. \quad (3)$$

Model recall was calculated using the following formula:

$$\frac{True\ Positive}{True\ Postive + False\ Negative}. \quad (4)$$

Model F1-score was calculated using the following formula:

$$2 \times \frac{Precision \times Recall}{Precision + Recall}. \quad (5)$$

**Feature visualization**. Feature visualization is a technique to gain insight into the potential features that the CNN leverages to predict classes. For feature visualization, we visualized the raw activation of the ReLU layer of the third convolutional module (i.e., ReLU 3) through a 3D reconstruction activation mapping technique. In this case, we used a preprocessed participant datum (i.e., segmented, smoothed, slice extracted, age-regressed image) and calculated the sum activation (using MATLAB's "activation" function) of all the 32 convoluted images of ReLU 3 of one trained network. We then min-max normalized the cumulative activation to obtain a normalized cumulative activation matrix for that datum. This was repeated for every preprocessed participant datum, each time for one of the (one hundred) fully trained CNN models. Depending on the initial brain slice (i.e., from −29 to +28 mm) and the disease group (i.e., TLE, AD, or healthy), we reconstructed an average 3-D topographic map (or disease activation brain) of the activations for each disease. To increase interpretability of the results and further elucidate the mechanism of our CNN, we again min-max renormalized each slice in our disease activation brains and extracted voxels in our reconstructed brain with values >0.75, setting the rest to 0. Finally, to examine the anatomical location of our high activation voxels, we used MRIcroGL to superimpose activation maps on the regions of interest (ROIs) derived from the Automated Anatomical Labeling (AAL) atlas. We chose the AAL atlas due to its vast presence in epilepsy research[25–28] and the ease of anatomical interpretability. For participants with left-sided TLE, we mirrored the activation maps along the sagittal axis and combined the maps with right-sided TLE participants to allow ipsilateral versus contralateral (relative to side of diagnosis) ROI analysis.

**Voxel-based morphometry**. To examine focal differences between disease groups and to provide a comparison for visualization among feature weight analyses and regions of group-wise brain atrophy, we employed voxel-based morphometry (VBM). We used independent samples $t$-test (two-tailed) comparisons between each disease group for each voxel in our preprocessed images, resulting in a total of three group comparisons (TLE vs. AD, TLE vs. Healthy, AD vs. Healthy). After accounting for multiple comparisons using Bonferroni corrected $p$-value threshold, we set the $t$-value of any voxel that was not significant to zero and min-max normalized the surviving values. Like our feature visualization, we extracted voxels in the reconstructed brain with values > 0.75, setting the rest to 0, and used MRIcroGL to examine which ROIs had high $t$-value voxels. For participants with left-sided TLE, we mirrored the $t$-value maps along the sagittal axis and combined the maps with right-sided TLE

participants to allow ipsilateral versus contralateral (relative to side of diagnosis) ROI analysis.

**Statistics and reproducibility**. All statistical calculations were calculated within MATLAB. VBM used two-tailed *t-test* with a Bonferroni corrected *p-value* threshold to identify statistically significant voxel intensities. Our sample size consisted of a total of 157 participants diagnosed with medial TLE, 73 participants diagnosed with early AD, and 251 healthy controls (150 derived from the epilepsy age-matched controls and 101 derived from the ADNI database). One hundred models were produced for the shuffled model, and one hundred models were produced for the properly trained model.

**Reporting summary**. Further information on research design is available in the Nature Portfolio Reporting Summary linked to this article.

## Results

**Disease prediction**. We trained 200 models (100 for the properly trained, 100 for shuffled) to predict disease using our pre-processed T1 image slices. Compared to the shuffled model, the trained model had a FDCI of 1 for age prediction (Fig. 5), i.e., it significantly outperformed the shuffled model 100% of runs, which is equivalent to observing these results by chance less than 1% of the time (i.e., $p < 0.01$). Trained models had a mean accuracy of 86.84% (SD = 1.33%), mean precision of 0.77

(SD = 0.03), a mean recall of 0.74 (SD = 0.03), and an F1-score of 0.75 (SD = 0.025) for group prediction. Our shuffled model had a mean accuracy of 67.16% (SD = 1.04%), mean precision of 0.41 (SD = 0.10), a mean recall of 0.33 (SD = 0.01), and an F1-score of 0.37 (SD = 0.1584) for group prediction. Disease dependent metrics for the trained models are displayed in Table 2.

**Feature visualization**. In our activation mapping, we examined the top average activations of the ReLU 3, the final layer before the classification layers, for all 100 of our trained models. The mean normalized activation for all AAL atlas regions in the reconstructed feature weight brain was 0.079 (SD = 0.09) for TLE, 0.083 (SD = 0.10) for AD, and 0.078 (SD = 0.08) for healthy controls. Reconstructed brain activation map and ROI activation matrices are displayed in Fig. 6.

**Voxel-based morphometry**. We used VBM techniques to examine focal differences in our preprocessed T1 images between disease types. The mean normalized *t*-value for all AAL atlas regions in the reconstructed VBM brain was 0.030 (SD = 0.06) for TLE versus healthy, and the results are displayed in Fig. 7.

## Discussion

In the current study, we investigated the use of CNN for TLE disease classification, with a special emphasis on discriminating TLE from another neurological disease with limbic atrophy (i.e., AD). We observed that a CNN model identified TLE vs. controls

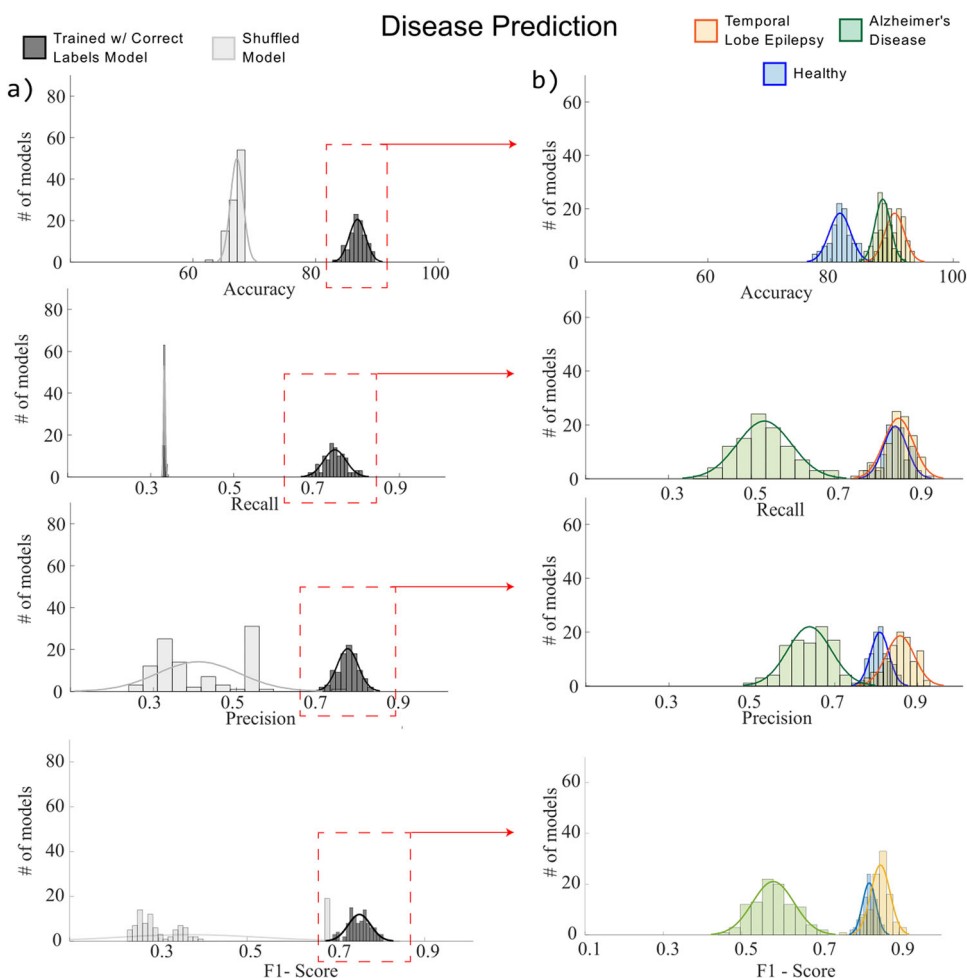

**Fig. 5 Model performance for disease prediction.** Model performance for disease prediction using our preprocessed brain slices. Column **a**: Gray bar graphs indicate correct label model performance (darker) and the shuffled models (lighter). Column **b**: Colored bars indicate disease specific performances.

**Table 2 Disease-dependent model evaluation metrics.**

|  | Accuracy (SD[a]) | Recall (SD) | Precision (SD) | F1-Score (SD) |
|---|---|---|---|---|
| Temporal lobe epilepsy | 90.45% (1.59%) | 0.86 (0.04) | 0.86 (0.03) | 0.85 (0.04) |
| Alzheimer's disease | 88.52% (1.27%) | 0.53 (0.07) | 0.64 (0.05) | 0.58 (0.05) |
| Healthy | 81.54% (1.77%) | 0.85 (0.03) | 0.81 (0.02) | 0.83 (0.02) |

[a]Standard deviation.

and TLE vs. AD at a statistically significant level independent of age. Importantly, the machine learning approach achieved high classification accuracy for TLE cases beyond the lesional status identified by human experts. Furthermore, our feature visualization analysis revealed that feature patterns were unique to each disease, highlighting temporal and extratemporal regions for TLE, leveraging traditional findings from VBM. Taken together, we demonstrated that CNN can accurately identify participants with TLE from AD and healthy controls while simultaneously revealing neural structures critical to the classification of each group. We discuss these findings in detail below.

The overarching aim of the current study was to test whether CNN could distinguish between epilepsy, AD, and healthy controls. As noted by our recent study that used machine learning for epilepsy classification[24], including another neurological disorder in addition to epilepsy (AD in the current study) was paramount, since it is possible that the CNN could be predicting non-specific anatomical changes related to the presence of a neurological disease involving limbic regions rather than the presence of TLE in particular. We chose AD because of the shared characteristics regarding temporal and limbic atrophy as well as global brain changes[29]. In other words, if we were to train a model to classify TLE, we would want to ensure that this model was not simply detecting temporal-limbic atrophy but, rather, TLE-specific changes. Indeed, we observed that the CNN was able to not only identify the presence of pathology but also accurately distinguish between the two diseases. These results mirror findings reported by other groups[30,31] and support the growing potential for clinical use of deep learning in diagnosis[32]. Since many deep learning models (including ours) leverage data that typically are a standard part of care for these diseases (e.g., MRI)[33], deep learning models potentially provide an inexpensive and effective method to aid clinicians in diagnosis by complementing human visual examination with other linear quantitative approaches. Furthermore, this would be especially beneficial to non-lesional cases underscored by the observation that our model was able to accurately predict TLE diagnosis (~90%) compared to only 47% by clinicians on MRI alone.

The performance of our CNN model and the results from the subsequent feature visualization analysis indicate that CNN leveraged whole brain TLE associated changes.

Global patterns of gray matter atrophy in TLE are supported by our CNN model, which used each individual's 58 axial slices (−29 to +28 mm) for training and testing. In other words, our CNN can accurately distinguish an axial slice taken from a TLE brain versus an axial slice taken from an AD brain. Since each of the 58 axial slices were tested independently to predict disease, specific patterns of gray matter unique to each disease (i.e., pathology) must have been present on most axial slices tested. This feature of our design is important because it ensures that disease classification is not reliant on a spurious finding of a single slice chosen arbitrarily for model training or on anatomical changes that are too local and could be missed by other unseen slices. Furthermore, our feature visualization analyses revealed broad disease-dependent global patterns highlighting areas that the CNN used for disease prediction. We interpret these areas as likely patterns

of pathology. Indeed, temporal-limbic areas that are well-documented as associated with TLE pathology (e.g., hippocampus, amygdala, parahippocampal, caudate, putamen, cingulum, and thalamus) were also highlighted by our feature visualization technique. However, extratemporal areas including those found in the frontal (e.g., orbital, and olfactory) and occipital cortices (e.g., precuneus), were also identified by our feature analysis, supporting quickly growing evidence that broad features of epilepsy are present beyond the temporal-limbic areas[24,34–36]. Lastly, ROIs such as the thalamus were highlighted by our feature analysis but not our VBM analysis, potentially introducing new insights and techniques to probing regions affected by TLE.

There are several clinical as well as research implications of the current study. From a clinical standpoint, CNN continues to demonstrate its role as a useful tool for aiding clinicians in the diagnosis of epilepsy[37]. As discussed above, the visual confirmation of a lesion on preoperative MRI has been consistently deemed a reliable marker of higher chances for post-surgical seizure freedom[11]. An important finding of the current approach is the superior detection of TLE itself even among cases that were deemed to be non-lesional in nature by human experts. In this study, we included medial temporal lobe cases whose structural MRI studies had been classified as either unremarkable (i.e., non-lesional) or as having obvious radiographic features of underlying hippocampal sclerosis pathology (i.e., lesional). 47% of patients fell into the latter group. Partly, this is due to the fact that TLE diagnosis by clinicians relies heavily on abnormalities in the hippocampus despite the known prominence of aberrant patterns in extra hippocampal regions both ipsi- and contralaterally in the brains of patients with unilateral focal epilepsy[24]. CNN overcomes this overreliance on hippocampal imaging findings through examination of subtle, diffuse disease-related pathology that may otherwise go undetected by human visual examination[38] (i.e., what is currently considered "non-lesional" epilepsy). Indeed, the CNN model's ~94% average accuracy in detecting TLE suggests that the machine learning approach can identify lesional patterns that are invisible to the human eye in a far larger number of cases. This is critical to the future implementation of AI-based tools in clinical settings because it suggests that the definition of lesional epilepsy may benefit from a human-machine synergistic interpretation and could change practice in many cases. For example, in many cases, if a patient were classified by the human experts as having "non-lesional" MRI imaging with unremarkable or equivocal PET but semiological and neurophysiological evidence suggestive of medial temporal onset, invasive EEG monitoring may be required before offering epilepsy surgery to confirm the area of seizure onset. However, if AI could reliably identify the presence of TLE pathology, lateralize it, and localize it, this could prompt the redefinition of "lesional status" and aid in the decision making of clinicians by integrating additional information to clinical, radiographic, and neurophysiological data. This could hence be cost-effective but also, more importantly, less invasive on patients desperately needing surgical intervention to achieve control of their seizures. Naturally, we do not propose that this single study should make such paradigmatic change; rather, we believe that the findings of this study provide

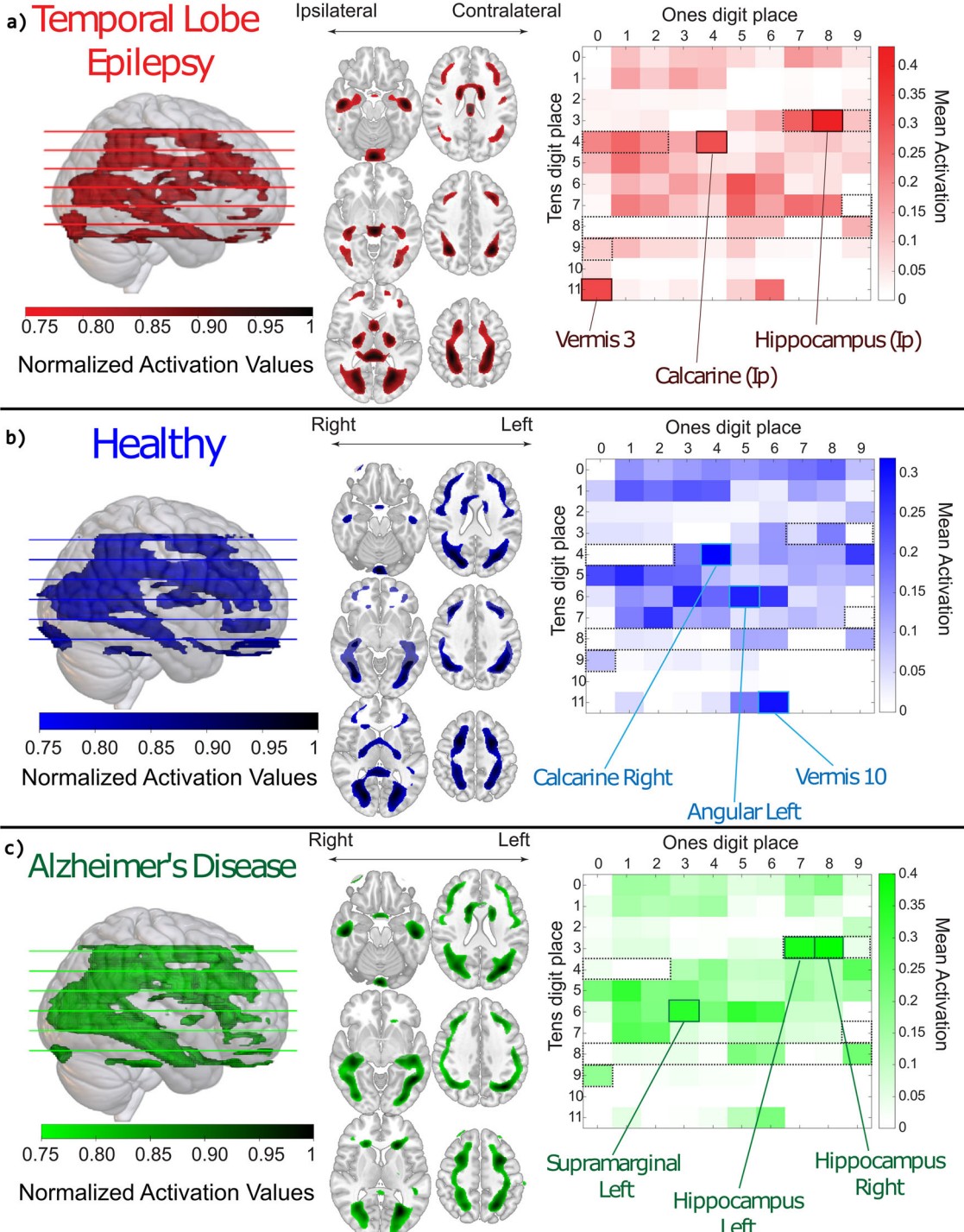

**Fig. 6 Feature visualization analysis of convolutional neural networks for disease prediction.** Analysis for each disease prediction is display in rows (**a**: TLE, **b**: Healthy, **c**: AD). Each row has a 3D/mosaic view of our activation mapping reconstruction with a horizontal color bar indicating normalized activation values for each slice and direction arrows either indicating left and right or for TLE, the diagnosis laterality, as well as a region of analysis (ROI) activation matrix using the Automated Anatomical Labeling (AAL) atlas. The vertical color bar indicates mean activation values within each ROI. Cells within the dotted lines indicate temporal ROIs. Ip Ipsilateral (relative to side of diagnosis), Con Contralateral.

an encouraging step towards the development of a machine learning pipeline that can yield the outputs necessary for translation into clinical settings. Timely, reliable, and accurate diagnosis is key in the treatment planning of epilepsy, including likelihood of surgical success. Hence, the development and further refinement of machine learning models for these purposes has potential major implications for the care of patients with

epilepsy. Taken together, we propose that our results could have a direct impact on clinical treatment and further our understanding of neuropathological changes in epilepsy. Our findings corroborate previous work on machine learning for the detection of both pediatric[39] and adult epilepsy[24], particularly TLE[37]. Overall, we build upon this work through showing that CNN can be disease-specific without the confounding effect of age.

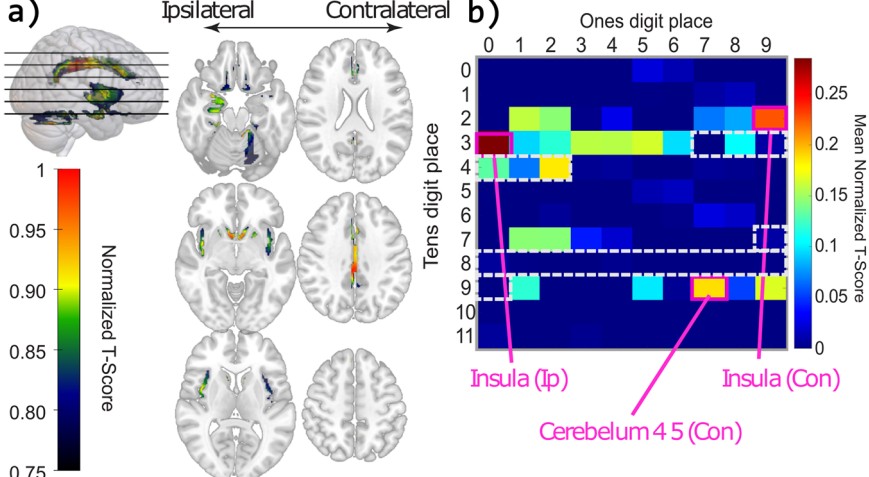

**Fig. 7 Voxel-based morphometry analysis for temporal lobe epilepsy vs health control.** For participants with left-sided TLE, we mirrored the maps along the sagittal axis and combined the maps with right-sided TLE participants. **a** Mosaic displays normalized t-values from statistically significant voxels. **b** Region of interest (ROI) matrix produced using the Automated Anatomical Labeling atlas. The color bar indicates mean normalized t-values within each ROI. Cells within the dotted lines indicate temporal ROIs.

The findings of the current study must be interpreted within the context of certain limitations. First, our study is restricted to a specific form of focal epilepsy and cannot be generalized to all epilepsy syndromes. It will be crucial for future cohorts to include not only extratemporal epilepsy but also probe whether the classifier can distinguish TLE from generalized genetic epilepsy (GGE) syndromes. This latter point is important, since GGE typically is deemed to be associated with a normal MRI brain and hence "non-lesional" in nature. Again, the possibility of a subtle pattern not otherwise detectable by human visual examination may redefine our understanding of lesions in epilepsy. In fact, preliminary evidence suggests a pattern of aberrant structural organization in GGE[40]. Second, while we identified and addressed the role of age as a potential confounder given our disease groups, it will be crucial for future studies to probe the role of other important confounding factors such as exposure to specific medications, years of education, etc. Finally, the approach proposed in this study could be extended to other applications within epilepsy as well as other disease models. For example, for challenging MRI-negative cases where the neurophysiological and semiological data suggest either GGE or a focal epilepsy with rapid bisynchrony, this approach may help distinguish the epilepsy syndrome. Naturally, lateralization of seizure foci is an important application of machine learning models and may help improve access for surgical evaluation of patients whose initial clinical and imaging data fail to provide a clearly lateralizing syndrome. Future work could use similar CNN approaches for a variety of outcomes is also of interest, ranging from response to antiseizure medications, dietary treatments, neuromodulation, and postoperative seizure outcome, although this study does not propose its efficacy at predicting these phenotypes. Extending this approach to other diagnostic challenges is also promising, including distinguishing Parkinsonian syndromes or different subtypes of vascular dementia as well as gauging the neuroanatomical signatures of conditions such as traumatic brain injury, among others. Additionally, future work could employ additional neuropsychological tests to examine MCI, dementia, and early, mid, and late-stage Alzheimer's disease. Thirdly, since we did not employ any exhaustive hyperparameter search, our model is not fully optimized. Although this is beyond the scope of this study, this is a limitation. Future studies can aim to optimize model through a hyperparameter search via techniques such as grid search, Bayesian, etc. to investigate the ideal hyperparameters such as learning rate, batch size, epoch length, etc. for this applied model as well as for the field of deep learning in medical imaging.

## Data availability

The complete datasets analyzed in the current study are not publicly available due to patient confidentiality restrictions set forth by the IRB. The datasets are available from the corresponding author upon reasonable request which may require a Data Use Agreement. The numerical data underlying the Figures can be found in Supplementary Data 1.

## Code availability

Custom code can be found on the following link: https://doi.org/10.5281/zenodo.7621235[41].

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

## Acknowledgements
Data collection and sharing for this project was funded by National Institute of Health (NIH) grants R01 NS110347 (PI Bonilha) and R01 NS088748 (PI Drane) as well as the Alzheimer's Disease Neuroimaging Initiative (ADNI) (National Institutes of Health Grant U01 AG024904) and DOD ADNI (Department of Defense award number W81XWH-12-2-0012). ADNI is funded by the National Institute on Aging, the National Institute of Biomedical Imaging and Bioengineering, and through generous contributions from the following: AbbVie, Alzheimer's Association; Alzheimer's Drug Discovery Foundation; Araclon Biotech; BioClinica, Inc.; Biogen; Bristol-Myers Squibb Company; CereSpir, Inc.; Cogstate; Eisai Inc.; Elan Pharmaceuticals, Inc.; Eli Lilly and Company; EuroImmun; F. Hoffmann-La Roche Ltd and its affiliated company Genentech, Inc.; Fujirebio; GE Healthcare; IXICO Ltd.; Janssen Alzheimer Immunotherapy Research & Development, LLC.; Johnson & Johnson Pharmaceutical Research & Development LLC.; Lumosity; Lundbeck; Merck & Co., Inc.; Meso Scale Diagnostics, LLC.; NeuroRx Research; Neurotrack Technologies; Novartis Pharmaceuticals Corporation; Pfizer Inc.; Piramal Imaging; Servier; Takeda Pharmaceutical Company; and Transition Therapeutics. The Canadian Institutes of Health Research is providing funds to support ADNI clinical sites in Canada. Private sector contributions are facilitated by the Foundation for the National Institutes of Health (www.fnih.org). The grantee organization is the Northern California Institute for Research and Education, and the study is coordinated by the Alzheimer's Therapeutic Research Institute at the University of Southern California. ADNI data are disseminated by the Laboratory for Neuro Imaging at the University of Southern California.

Data used in preparation of this article were obtained from the Alzheimer's Disease Neuroimaging Initiative (ADNI) database (adni.loni.usc.edu). As such, the investigators within the ADNI contributed to the design and implementation of ADNI and/or provided data but did not participate in analysis or writing of this report. A complete listing of ADNI investigators can be found at: http://adni.loni.usc.edu/wp-content/uploads/how_to_apply/ADNI_Acknowledgement_List.pdf.

## Author contributions
A.J.C. and R.R. developed manuscript. A.J.C., E.B., and J.W. analyzed data. T.R., S.K., D.D., R.G., A.A., V.C., I.K., E.K., B.W., and C.M. contributed to the development of experiment and manuscript. E.G. and L.B. supervised this experiment.

## Competing interests
The authors declare no competing interests.

## Additional information

# Alzheimer's Disease Neuroimaging Initiative

Leonardo Bonilha[2,10]✉

A full list of members and their affiliations appears in the Supplementary Information.

