## [Peer Review File · Communications Medicine]

Reviewers' comments:

Reviewer #1 (Remarks to the Author):

In this manuscript, the authors applied CNN-based deep learning to classification between TLE, AD, and HCs, using structural MRI. They obtained brain MRI data of these cohorts from their institutes as well as public database and analyzed them by SPM/CAT and CNN algorithm. As a result, their model achieved generally >80% accuracy and precision in terms of classifying TLE, AD, and HC.

It is an interesting experiment, and the manuscript is generally well-written. I have some comments below on this article.

- In addition to data on Table 1, more detailed clinical information is needed, including MMSE and CDR for AD, onset age and seizure-related data for TLE. With such information, readers consider about further application for the real patients.

- How did the authors define early AD and differentiate them from MCI? Please describe more details.

- 2.2.4. voxel-wise linear age regression. I think the validity of this process is quite important for this study. Age-related GM changes are not always linear, and there should be scanner difference as well. Probably ADNI data are mostly from 1.5T scanner. Therefore, I would suggest an additional analysis to validate this correction method for age. If there is no difference in voxel-wise comparison between HC from TLE cohort and HC from ADNI, this age correction should have worked well. If there's significant difference between both HC groups, there should be some bias.

Reviewer #2 (Remarks to the Author):

This is a novel work wherein the authors have assessed a convolutional neural network (CNN) algorithm to classify TLE vs. healthy controls vs. patients with AD using T1-weighted magnetic resonance imaging (MRI) scans. They have used feature visualization techniques to identify regions the CNN employed to differentiate disease types. Model predictions were also significantly better than random permutation classifications ($p < 0.01$) and were independent of age effects. Feature weight CNN activation was disease-dependent and included temporal and extra-temporal regions commonly associated with TLE, demonstrating the importance of a whole-brain approach.

The following issues are my observations require authors attention.

1. Change title of the paper appropriately.
2. Organization of the paper is very poor. Need a better structure.

For example, section 2. Methods

2.1: Participants

Put some content for section subsection. Then start 2.1.1.

3. Epoch size is very small to obtain a generalized model. Need to use 1000 epoch or more and find at which particular epoch your loss function is minimum and remain constant for

remaining epochs.

4. Though it is a sensitive problem, we cannot compromise with False Positive and True Negative. So, instead of using the default hyper parameter values, I would suggest to use an optimization algorithm like Bayesian algorithm to enhance the efficiency of the classification.

5. Evaluation metrics are not dependable as the problem is a multi-class problem. Compute F1-score metrics.

6. Evaluated results must be provided in a tabular form instead of a paragraph to improve readability of the manuscript.

7. Formatting and grammatical mistakes must be taken care of by the authors.

Reviewer #3 (Remarks to the Author):

The authors report their experience about the possibility to classify and distinguish Temporal Lobe Epilepsy from Alzheimer's disease. Their data are interesting and novel and will be probably of great interest to other researchers in the scientific community. In my opinion, the topic is important and it is often poorly understood in the clinical practice.

I have few comments for the authors:

- Introduction: 4^o line: I suggest to change reference 3 with the newer reference: Scheffer IE, Berkovic S, Capovilla G et al. *Epilepsia*. 2017;58:512-21.

In general, this section contains many well-known information; it is advisable to summarize the text, focusing on the topic of the study.

- Data Sources: It is not mentioned during which period of time the collection of the patients was done; this should be added; moreover, is the selection of the patients in any way systematic or random? Later, the authors state about "scalp EEG patterns" but they do not describe these patterns: I suggest to describe in detail the EEG abnormalities of the patients studied.

- Training/Validation/Testing Set Allocation: Who performed group allocation?

- Slice-Wise 2D CNN Accurately Predicts Disease Type: In the last lines of the text, the authors state about a possible aid for the clinicians in diagnosis of temporal lobe epilepsy but, in my opinion, it is unclear in what type of patient can be useful this technique; I suggest to specify better this possible aid.

- Clinical and Theoretical Implications: Again, for me it is unclear if clinical implications concern all patients with temporal lobe epilepsy or just a subgroup of these patients (e.g. MRI-negative cases): this point is crucial because often diagnosis of temporal lobe epilepsy is clear and I cannot understand the reason to perform CNN in all patients. I think that this point should be clarified. Finally, the authors suggest a possible role in the evaluation of antiseizure medication treatment: please, describe in a more incisive way.

Reviewer #1

In this manuscript, the authors applied CNN-based deep learning to classification between TLE, AD, and HCs, using structural MRI. They obtained brain MRI data of these cohorts from their institutes as well as public database and analyzed them by SPM/CAT and CNN algorithm. As a result, their model achieved generally >80% accuracy and precision in terms of classifying TLE, AD, and HC. It is an interesting experiment, and the manuscript is generally well-written. I have some comments below on this article.

Thank you for the kind words and constructive comments about our work. We believe the manuscript has been greatly improved based on your suggestions.

In addition to data on Table 1, more detailed clinical information is needed, including MMSE and CDR for AD, onset age and seizure-related data for TLE. With such information, readers consider about further application for the real patients.

We agree with the reviewer that providing available sociodemographic and clinical characteristics of our patient cohorts can improve the interpretation of the main findings. We have accordingly added age at onset and duration of disease for the TLE group to provide the readers a more comprehensive clinical depiction of the epilepsy patients studied. Unfortunately, we do not have access to instruments such as the Mini Mental State Examination or Clinical Dementia Rating as our AD data were obtained through ADNI, which only includes diagnosis, age, and gender in its basic neuroimaging dataset. However, we can confirm that inclusion criteria for ADNI stipulated an abnormal memory function score on Wechsler Memory Scale; Mini-Mental State Exam (MMSE) score of 20-26; and a minimum Clinical Dementia Rating (CDR) = 0.5 (<https://www.alzheimers.gov/clinical-trials/alzheimers-disease-neuroimaging-initiative-2-adni2>) (ref <https://www.ncbi.nlm.nih.gov/pmc/articles/PMC2809036/>)

How did the authors define early AD and differentiate them from MCI? Please describe more details.

We relied on ADNI labeling for AD versus MCI diagnosis. ADNI diagnoses AD based on the National Institute on Aging-Alzheimer's Association (NIAAA) guidelines for the neuropathological assessment of AD. As such, this is a gold standard stratification which is used by world experts in dementia. It is standard procedure to assume the diagnostic classification by ADNI is valid. Further details on the inclusion and classification criteria of the ADNI study can be found at: <https://www.alzheimers.gov/clinical-trials/alzheimers-disease-neuroimaging-initiative-2-adni2>. More specifically, as indicated in Bondi et. al. (ref <https://www.ncbi.nlm.nih.gov/pmc/articles/PMC4133291/>), the ADNI criteria for MCI required the following characteristics: 1) subjective memory complaints; 2) a score lower than the education adjusted normal lower limit cut-off of the delayed recall of Story A of the Wechsler Memory Scale Revised (WMS-R) Logical Memory Test; 3) global CDR score of 0.5; and 4) a physician ascertainment of sufficiently preserved functional performance. In turn, the diagnosis of early AD required: an MMSE score between 20-26, CDR of 0.5 or 1.0, and meeting NINCDS/ADRDA criteria for probable AD (https://adni.loni.usc.edu/wp-content/themes/freshnews-dev-v2/documents/clinical/ADNI-1_Protocol.pdf).

2.2.4. voxel-wise linear age regression. I think the validity of this process is quite important for this study. Age-related GM changes are not always linear, and there should be scanner difference as well. Probably ADNI data are mostly from 1.5T scanner. Therefore, I would suggest an additional analysis

to validate this correction method for age. If there is no difference in voxel-wise comparison between HC from TLE cohort and HC from ADNI, this age correction should have worked well. If there's significant difference between both HC groups, there should be some bias.

We agree that the validity of our process for accounting for age is pivotal. Encouraged by the reviewer's suggestion, we performed additional voxel-wise analyses to test whether age related tissue atrophy would be better explained by linear vs. non-linear models (e.g., exponential, and power regression). Our goal was to compare whether 1) the relationship between anatomical sites of maximal age-related changes would be different depending on the statistical approach; and 2) which model best captured the explained variance in tissue volume based on age. Therefore, we performed new statistical analyses where the voxel-based tissue was regressed or correlated with age. The results are summarized below. On the top row, we show R^2 maps for linear, exponential, and power regression. The bottom row displays a histogram of the R^2 values for linear, exponential, and power regression. Overall, we observed that the models were consistent with regards to the locations where atrophy was associated with age, i.e., there were no anatomical discrepancies between the models. Moreover, as demonstrated by the voxel-wise maps and histograms, the linear regression outperformed both non-linear models in terms of the variance explained. Therefore, we believe that this is an empirical demonstration that a linear age-tissue volume model is an adequate approach to capture the influence of age, and our adopted approach to control for age by measuring the residual value between age and tissue volume can best take into account the age-related atrophy.

Reviewer #2

This is a novel work wherein the authors have assessed a convolutional neural network (CNN) algorithm to classify TLE vs. healthy controls vs. patients with AD using T1-weighted magnetic resonance imaging (MRI) scans. They have used feature visualization techniques to identify regions the CNN employed to differentiate disease types. Model predictions were also significantly better than random permutation classifications ($p < 0.01$) and were independent of age effects. Feature weight CNN activation was disease-dependent and included temporal and extra-temporal regions commonly associated with TLE, demonstrating the importance of a whole-brain approach. The following issues are my observations require authors attention.

Thank you for the helpful and encouraging comments. We appreciate your suggestions, and we believe that our revised manuscript has been strengthened by your feedback.

Change title of the paper appropriately.

We agree that the title was not adequately depicting the motivation and findings of this study. We have now changed it to:

MRI-based deep learning can discriminate between Temporal Lobe Epilepsy, Alzheimer's disease, and healthy controls.

Organization of the paper is very poor. Need a better structure. For example, section 2, methods (2.1: Participants). Put some content for section subsection. Then start 2.1.1.

We regret that the paper organization was suboptimal in the initial submission. We have reformatted the manuscript to reflect a more coherent flow. We have completely revised the study to incorporate the suggestions and to include sections and subsections as recommended.

Epoch size is very small to obtain a generalized model. Need to use 1000 epoch or more and find at which particular epoch your loss function is minimum and remain constant for remaining epochs.

Although we do agree with the reviewer that, a priori, an epoch size of 30 could be small, we empirically observed that 30 epochs were sufficient to obtain a minimum loss function relatively quickly (~10 epochs), which remained constant for the remaining epochs. There is no universal consensus on the number of epochs that is optimal, and hence, we elected to choose one that demonstrated validity from a practical standpoint. To demonstrate this visually, we enclose below a sample of the training progress MATLAB outputs for an example model. Please note the convergence to plateaued accuracy and minimal loss at approximately 30 epochs. We agree with the reviewer that this is a topic for further explorations as the field of deep learning in medical images continues to expand, and we have included this point in the discussion of our manuscript accordingly.

Though it is a sensitive problem, we cannot compromise with False Positive and True Negative. So, instead of using the default hyper parameter values, I would suggest to use an optimization algorithm like Bayesian algorithm to enhance the efficiency of the classification.

This is a very important and insightful comment, and we believe that we needed to better explain our rationale for hyperparameter choices. More specifically, we aimed to demonstrate the accuracy of deep learning even with default parameters. We certainly agree that hyperparameter optimization can be useful to refine the most efficient and optimal models. Nonetheless, as the reviewer is aware, parameter optimization can be a highly demanding and time intensive task. Even though optimization techniques like Bayesian algorithms can help alleviate this process, there is no consensus on the best approach is (i.e., manual, grid, Bayesian, or etc.). More specifically, this is the first study, to the best of our knowledge, which is capable of distinguishing two pathological cohorts with shared neuroanatomical substrates not only from healthy controls but from each other. And here, we demonstrate that convolutional neural networks can leverage the well-known subtle voxel-wise patterns of atrophy when using tissue maps. Indeed, albeit invisible to the human eye, tissue atrophy patterns have been consistently demonstrated in the literature but have not been leveraged for translational purposes. Thus, by demonstrating the high accuracy of classification with the default parameters, we demonstrate the utility and impact of the approach of using tissue maps for epilepsy and dementia classification, which is an important reconciliation with decades of voxel-wise analyses in these fields.

Nonetheless, we believe that the reviewer raises and central point to the field of AI, neuroimaging, and medicine. There are several opportunities for model refinement that have not been explored in this paper (e.g., hyperparameter grid or Bayesian searches, raw image input, three-dimensional anatomically informed [limbic] models). Naturally, an exhaustive treatment on all these variables is beyond the scope of this study, but we acknowledge this important point, and we have revised our manuscript to include these points in the discussion and limitations.

Evaluation metrics are not dependable as the problem is a multi-class problem. Compute F1-score metrics.

We agree with the reviewer and have, accordingly, calculated and added F1 score metrics to our results section and figure (see below and on the manuscript).

Evaluated results must be provided in a tabular form instead of a paragraph to improve readability of the manuscript.

We have added table 2 to the main text, which reports accuracy, recall, precision, and F1 score for each group in tabular form. We have included this table here for ease of access to the reviewer:

	Accuracy	Recall	Precision	F1-Score
Temporal Lobe Epilepsy	90.45% (1.59%)	0.86 (0.04)	0.86 (0.03)	0.85 (0.04)
Alzheimer's disease	88.52% (1.27%)	0.53 (0.07)	0.64 (0.05)	0.58 (0.05)
Healthy	81.54% (1.77%)	0.85 (0.03)	0.81 (0.02)	0.83 (0.02)

Formatting and grammatical mistakes must be taken care of by the authors.

Thank you for this comment. We have reviewed the manuscript thoroughly to correct any pending typos or errors.

Reviewer #3

The authors report their experience about the possibility to classify and distinguish Temporal Lobe Epilepsy from Alzheimer's disease. Their data are interesting and novel and will be probably of great interest to other researchers in the scientific community. In my opinion, the topic is important and it is often poorly understood in the clinical practice. I have few comments for the authors:

Thank you for the encouraging comments and constructive remarks.

Introduction: 4th line: I suggest to change reference 3 with the newer reference: Scheffer IE, Berkovic S, Capovilla G et al. Epilepsia. 2017;58:512-21

Thank you for indicating this reference, and we have thus changed it to the newer reference.

In general, this section contains many well-known information; it is advisable to summarize the text, focusing on the topic of the study.

We have shortened the introduction to make the goals of our manuscript clearer.

Data Sources: It is not mentioned during which period of time the collection of the patients was done; this should be added; moreover, is the selection of the patients in any way systematic or random? Later, the authors state about "scalp EEG patterns" but they do not describe these patterns: I suggest to describe in detail the EEG abnormalities of the patients studied.

We have added the following clarification to the main text:

“Patients were recruited sequentially between March 2017 and December 2020 if they met the following inclusion criteria: 1) a diagnosis of drug-resistant unilateral temporal lobe epilepsy was achieved by the treating clinical team based on a combination of clinical, neurophysiological, radiographic, and neuropsychological findings in accordance with criteria set for by the International League Against Epilepsy (ILAE)³⁴; 2) treatment with either resection or laser interstitial thermoablation as decided by the patient and his/her clinical team; 3) age > 18 years old. Patients were excluded if they had mass occupying lesions (e.g. tumors, vascular malformations), as these tend to distort the anatomy, if they did not undergo resective/ablative surgery, or if they were found to have bilateral temporal lobe epilepsy or an additional extratemporal focus.”

Training/Validation/Testing Set Allocation: Who performed group allocation?

The group allocation was performed using a randomization approach implemented as part of the deep learning workflow in MATLAB. Specifically, we used the “dividerand” MATLAB function ([mathworks.com/help/deeplearning/ref/dividerand.html](https://www.mathworks.com/help/deeplearning/ref/dividerand.html)). We chose this approach as it provides a systematic way to randomly allocate participants to each cohort without human bias. To make this clear, we have added the following to the main text.

“Randomized group allocation was performed before each model construction and evaluation using MATLAB’s “dividerand” function (<https://www.mathworks.com/help/deeplearning/ref/dividerand.html>).”

Slice-Wise 2D CNN Accurately Predicts Disease Type: In the last lines of the text, the authors state about a possible aid for the clinicians in diagnosis of temporal lobe epilepsy but, in my opinion, it is unclear in what type of patient can be useful this technique; I suggest to specify better this possible aid.

This is a very important point, so we appreciate the opportunity to delve into this further. Briefly, our results show a high accuracy in TLE diagnosis of ~90%. Notably, however, only 47% of the TLE cohort was deemed lesional based on MRI alone. This classification of lesional vs. non-lesional was derived from the clinical consensus of each patient's treating team at each site. This means that a group of human experts who typically have access to T1, T2, FLAIR and other sequences across different orientations (coronal, axial, sagittal, oblique) in addition to clinical and neurophysiological information only deemed about half of the patients as having MRI findings of TLE. By showing an accuracy of ~90%, our deep learning model is capable of detecting something from the axial T1-weighted slices alone, with no other information. This suggests that AI-based models could aid in TLE diagnosis by merely identifying these patients when human experts fail to identify overt signs of TLE on MRI (e.g., hippocampal sclerosis). We have added the following to the main text

“In conclusion, AI (CNN deep learning) can successfully classify and distinguish TLE, underscoring its potential utility for future computer-aided radiological assessments of epilepsy, especially in patients who do not exhibit overtly identifiable TLE-associated MRI features (e.g., hippocampal sclerosis).”

Clinical and Theoretical Implications: Again, for me it is unclear if clinical implications concern all patients with temporal lobe epilepsy or just a subgroup of these patients (e.g. MRI-negative cases): this point is crucial because often diagnosis of temporal lobe epilepsy is clear and I cannot understand the reason to perform CNN in all patients. I think that this point should be clarified.

This is a very important, and we believe that it merits further clarification on our part. As the reviewer is aware, hippocampal atrophy does not equate temporal lobe epilepsy. Conversely, normal hippocampal appearance on MRI does not exclude the diagnosis of temporal lobe epilepsy. Therefore, there is a clinical (and mechanistic) need for the identification of a model that can abridge whole brain information that is invisible to the human eye but can be used to classify temporal lobe epilepsy based on a limbic pattern of brain atrophy. Naturally, the clinical context in which this approach would lead to maximal yield would be in cases of non-lesional (normal MRI) temporal lobe epilepsy, or in cases of hippocampal atrophy where there are discordant information and the differential diagnosis need to be refined.

For these reasons, this study included patients with and without hippocampal atrophy. This is first study of its kind in the neuroradiology of epilepsy. As the reviewer appreciated, the anatomical regions “driving the diagnosis of temporal lobe epilepsy” were indeed partly located in the hippocampus. But not exclusively, thus corroborating the notion that visual inspection alone of a hippocampal atrophy case can miss the presence or absence of these markers. A more critical confirmation of this point is the Alzheimer's disease cohort, which has pronounced hippocampal atrophy and whose hippocampal atrophy also plays a role in diagnosis, but the overall brain and temporal lobe (medial and lateral) patterns can discriminate between the disease. Indeed, Alzheimer's disease was chosen as the comparative cohort in this study on purpose for this very reason.

Therefore, our study does suggest that a predictive model is possible for the identification of lesional and non-lesional temporal lobe epilepsy. Theoretically, this further contributes to the idea of focal epilepsy as a disease of networks, and it leverages those networks for clinical applicability (see below). Second, a major novelty of this study is the demonstration that AI-based models are not driven primarily by the overt area of pathology. As the reviewer knows, both TLE and Alzheimer disease show primary and prominent (although certainly not exclusive) atrophy in the mesial temporal regions. If deep learning model were only “paying attention” the mesial temporal lobe (as humans do), then it would be unable to tease apart TLE from AD. Yet, despite removing the effects driven by age, our algorithm can discriminate between these diseases. We think that a novelty of this study is introducing this framework to the clinical/translational literature, which may be useful to model future other studies. Finally, in terms of clinical applications, the fact that the machine can classify a patient as TLE with such high accuracy could redefine the role of “lesional” status in the future. Here we clarify that we do not propose that this single proof-of-concept study should change this paradigm, but we propose that there is value in this approach. For example, on non-lesional patients, deep learning can leverage markers invisible to the human eye to significantly improve the differential diagnosis.

We agree that this is an important point which merits clarification, notably with the explanation that the identification of temporal lobe epilepsy markers from lesional and non-lesional cases can provide improved accuracy for diagnosis and each group, and eventually non-lesional cases, are likely to best benefit from this approach after the conceptual demonstration has been established for both groups, as in this paper. We have expanded our text to include these important points as suggested.

Finally, the authors suggest a possible role in the evaluation of antiseizure medication treatment: please, describe in a more incisive way.

Thank you for bringing up this important point. Our work does not directly shed light on medication treatment, so we have clarified this as an avenue for future research using similar models. Naturally, one could consider that these methods could be used to further identify epilepsy phenotypes (e.g., related to treatment response). However, this was not tested here, and our comment was intended only to provide an outlook for future related work. We have clarified this accordingly.

REVIEWERS' COMMENTS:

Reviewer #1 (Remarks to the Author):

The authors sufficiently addressed my concerns. No further comments.

Reviewer #2 (Remarks to the Author):

Manuscript is now properly structured and the modifications suggested to the authors are now incorporated. Authors are advised to prepare it following the format of the journal.

Reviewer #3 (Remarks to the Author):

In my opinion, now the manuscript is really improved and delivers important messages. No other changes are requested.